# DCoM: A Deep Column Mapper for Semantic Data Type Detection

## Abstract

Detection of semantic data types is a very crucial task in data science for automated data cleaning, schema matching, data discovery, semantic data type normalization and sensitive data identification. Existing methods include regular expression-based or dictionary lookup-based methods that are not robust to dirty as well unseen data and are limited to a very less number of semantic data types to predict. Existing Machine Learning methods extract a large number of engineered features from data and build logistic regression, random forest or feedforward neural network for this purpose. In this paper, we introduce DCoM, a collection of multi-input NLP-based deep neural networks to detect semantic data types where instead of extracting a large number of features from the data, we feed the raw values of columns (or instances) to the model as texts. We train DCoM on 686,765 data columns extracted from the VizNet corpus with 78 different semantic data types. DCoM outperforms other contemporary results with a quite significant margin on the same dataset achieving a support-weighted F1 score of 0.925.

## 1 Introduction

Robust data preprocessing pipeline enables organizations to make data-driven business decisions. Most unstructured and semi-structured data after initial pre-processing is available in tabular format for further processing. These tabular datasets have to go through a process of data curation and quality check before consumption. Companies apply standard data quality checks/rules on business-critical columns to access and maintain the quality of data. Additionally, some of these datasets may contain sensitive information e.g. Protected Health Information (PHI), Personally Identifiable Information(PII), etc and need to be masked/de-identified i.e. identifying all columns which are Social Security Numbers, First Names, telephone numbers, etc. The first step in this data curation and quality check process is column level semantic tagging/mapping. There have been many attempts by Ramnandan et al. (2015), Limaye et al. (2010), Puranik (2012), Hulsebos et al. (2019), and Zhang et al. (2019) to automate this process. Traditionally, Semantic tagging is usually done using handcrafted rule-based systems e.g. tools from Google (2019), Microsoft (2019), etc. In some cases, column descriptions are also available. However, as the data grows exponentially, the cost associated with maintaining a rule-based system also increases. The semantic data type tagging is still largely manual where data stewards manually scan through databases and map columns of interest. Most ML-based approaches e.g. Zhang et al. (2019), Hulsebos et al. (2019) and Pham et al. (2016) use corresponding metadata, atomic datatypes, column description along with handcrafted features as input data for training the model.

We herein, propose **DCoM-D**eep **Co**lumn **M**apper; a generic collection of Deep learning based Semantic mappers to map the columns to semantic data types. These models take the raw values of the columns (or instances) as inputs considering those as texts and build NLP-based deep learning architectures to predict the semantic type of the columns (or instances). We also extracted 19 engineered features and used those as auxiliary features to DCoM models. We refrained from extracting a large number of features because we wanted to make the DCoM models learn those features along with some interesting high-level features on their own, for better semantic data type detection. As we are exploring NLP in this problem, we make the leverage of using advanced NLP-based deep learning layers or models such as Bi-LSTM, BERT, etc.

## 2 RELATED WORK

Microsoft (2019) Power BI and Google (2019) Data Studio use some regular expression based patterns for dictionary looks ups of column headers and values to detect a limited number of semantic data types. Venetis et al. (2011) build a database of value-type mappings, then assign semantic types using a maximum likelihood estimator based on column values. Syed et al. (2010) use column values and headers to build a Wikitology query to map columns to semantic classes. Ramnandan et al. (2015) use heuristics to first separate numerical and textual types, then describe those types using the Kolmogorov-Smirnov (K-S) test and Term Frequency-Inverse Document Frequency (TF-IDF), respectively. Pham et al. (2016) use slightly more features, including the Mann-Whitney test for numerical data and Jaccard similarity for textual data, to train logistic regression and random forest models. Goel et al. (2012) use conditional random fields to predict the semantic type of each value within a column, then combine these predictions into a prediction for the whole column. Limaye et al. (2010) use probabilistic graphical models to annotate values with entities, columns with types, and column pairs with relationships. Puranik (2012) proposes a "specialist approach" combining the predictions of regular expressions, dictionaries, and machine learning models. More recently, Yan & He (2018) introduced a system that, given a search keyword and set of positive examples, synthesizes type detection logic from open source GitHub repositories. Hulsebos et al. (2019) built a multi-input deep neural network model for detecting 78 semantic data types, extracting 686, 765 data columns from VizNet corpus from Hu et al. (2019). They extracted a total of 1,588 features for each column to train the model, thus resulting in a support-weighted F1 score of 0.89, exceeding that of machine learning baselines, dictionary and regular expression benchmarks, and the consensus of crowdsourced annotations. Zhang et al. (2019) introduced SATO, a hybrid machine learning model to automatically detect the semantic types of columns in tables, exploiting the signals from the context as well as the column values. It combines a deep learning model trained on a large-scale table corpus with topic modelling and structured prediction to achieve support-weighted and macro average F1 scores of 0.925 and 0.735, respectively, exceeding the state-of-the-art performance by a significant margin. Recently Deng et al. (2020) used a structure-aware Transformer encoder to capture semantics and knowledge in large-scale data. Wang et al. (2021) proposed Table Convolution Network and Suhara et al. (2021) proposed a multi-task learning approach (called Doduo) which outperform the existing results in semantic data type detection. Iida et al. (2021) devised a simple pre-training objective (corrupt cell detection) that learns exclusively from tabular data and reaches the state-of-the-art on a suite of table-based prediction tasks.

While in this paper we do not experiment with the context information of the column values, our work is mostly aligned to Hulsebos et al. (2019). Keeping that in mind, we are planning to research and implement the context of the column values in tables in our DCoM models for our future works. The paper is organized as follows: Section 3 describes the data used for DCoM models. In Section 4 we present the details of data preparation and architecture for DCoM models. Section 5 contains the training, evaluation and inference procedures of DCoM models while Section 6 discussed about the results of extensive experiments. In Section 7, we talk about some known limitations from the data as well as application point of view and finally, in Section 8, we present the concluding remarks and some future directions.

## 3 DATA

We have used the dataset prepared by Hulsebos et al. (2019) and compared our model performance with them considering their results to be baselines. This dataset contains 686,765 instances with 78 unique classes (or semantic data types). It is divided into 60/20/20 training/validation/testing splits. The instance count for classes varies from 584 (affiliate class) to 9088 (type class). The count distribution for classes is shown in Figure 2 of Appendix A. To provide a more clear picture of the data a sample of the dataset is also shown in Table 7 of Appendix A.

## 4 PROPOSED METHOD

Prior approaches (Venetis et al., 2011; Limaye et al., 2010) to semantic type detection trained models, such as logistic regression, decision trees, feedforward neural network (Hulsebos et al., 2019), extracting various features from the data. We, on the other hand, treated the data like natural

language (text) and used the data itself as the input to the model. We used a very small number of hand-engineered features unlike Hulsebos et al. (2019). Therefore, our DCoM models have two inputs, the values of the instances as text or natural language and hand-engineered features. We present two types of DCoM models based on the way we feed the text input to the model.

## 4.1 DCoM with Single Sequence Input

In this subsection, we discuss how the values of each instance can be fed to the model as a single sequence input to the DCoM. Considering the scenario, we can not simply pass the inputs separating values of each input with any separator token. This gives the model wrong information about the sequence of the data resulting in faulty training and poor performance on the unseen data. For example, if we consider the last example from Table 7 and create the input, `Deletes the property <SEP> Lets you edit the value of the property <SEP> Script execution will be stopped` for the model, the model gets to learn that the value `Lets you edit the value of the property` has a relative position between `Deletes the property` and `Script execution will be stopped`, which is quite wrong because the values in an instance do not have any relative position between them.

To mitigate the problem we introduce permutations from mathematics. With permutation, we order $r$ items from the set of $n_i$ items. Here $n_i$ items are all the values of an instance $i$, and $r \in [1, n]$. Doing so, the above instance can be broken down into multiple subsets. A sample of the subset is shown in Table 1. If we feed all the new instances of the subset to the model, it does not learn any relative positional information of `Lets you edit the value of the property` with respect to other values in that instance unlike earlier. Therefore, for the value `Deletes the property`, the model only learns the relative positions of the tokens e.g. `Deletes, the, property,` etc in a value, but not the relative position of the values in an instance. This helps the model getting the actual information present in the data for accurate prediction on the output. This permutation method also helps in augmenting new instances which help in training the data-hungry deep learning models with enriched data. It is not feasible to generate all the possible permutations before the training because of the huge number of subsets. Instead, during training, we sample $r$ between 1 and $n$ for each of the instances and generate one permutation for each instance. More than one instance can be generated, but training the model for multiple epochs will result same for both cases.

| New Instance | class |
|---|---|
| Deletes the property | description |
| Lets you edit the value of the property | description |
| Script execution will be stopped | description |
| Deletes the property <SEP> Lets you edit the value of the property | description |
| Lets you edit the value of the property <SEP> Script execution will be stopped | description |
| Deletes the property <SEP> Lets you edit the value of the property <SEP> Script execution will be stopped | description |
| Deletes the property <SEP> Script execution will be stopped <SEP> Lets you edit the value of the property | description |
| Lets you edit the value of the property <SEP> Deletes the property <SEP> Script execution will be stopped | description |

Table 1: Permutations on the values of a single instance from class `description`.

## 4.2 DCoM with Multiple Sequences Inputs

Input preparation with this method is straightforward compared to the earlier method. In this method, we also use permutations to generate new instances, but the value of $r$ is fixed during training and it is used as a hyper-parameter, where $r \in [1, \infty)$. Once we decide the value of $r$, $r$ number of inputs are used as text inputs to the model. Aggregation of embedding vectors of the inputs are performed once they are generated using the shared embedding weights. If $r > n$, where $n$ is the number of values of an instance, then this scenario can be handled in two ways. The first way is to pad $r - n$ inputs and aggregate the embedding vectors only for the non-padded inputs. Another way is while generating new instances use permutation with replacement to always sample $r$ values out of $n$ values. Therefore, padding is not required for the latter method. We tried both approaches and did not observe any significant difference in the result.

We used 19 out of 27 global statistical features from Hulsebos et al. (2019) as our engineered features for the DCoM models because we are only considering the unique values of columns. These features are normalized before feeding to the model. The complete list of these features is shown in Table 5. Once the text and engineered inputs are prepared, we attach LSTM/Transformer/BERT layers to the text input. The output of the earlier layers is aggregated with the engineered inputs. We use

some feed-forward layer after that. Finally, we use one `softmax` layer with 78 units to get the probability of each of the classes as output. This is our generalized architecture design of DCoM models. Extensive hyper-parameter tuning is performed to finalize the number of layers, number of units in a layer and many other hyper-parameters in the model. This topic is discussed in detail in section 5. To name the DCoM models, the type of the text input and the name of the deep learning layers are used as suffixes with the name `DCoM`, e.g. DCoM model with single instance input and LSTM layers are named as `DCoM-Single-LSTM`. The architecture design of DCoM models for both single sequence and multiple sequences inputs are shown in Figure 1.

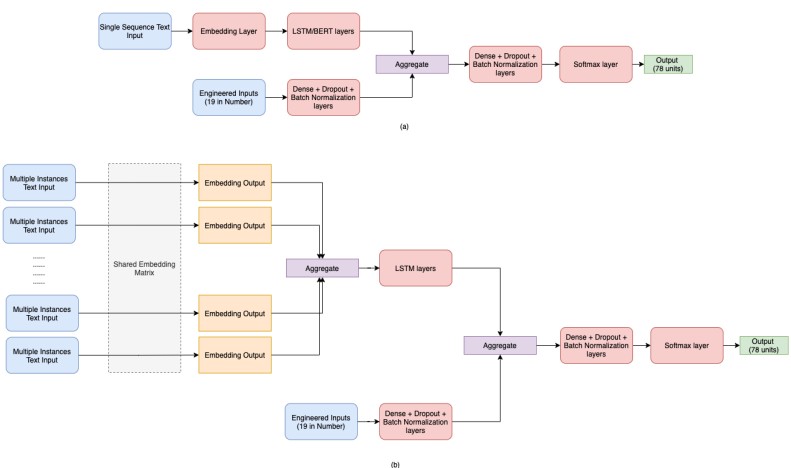

Figure 1: Architecture diagram of (a) `DCoM-Single` and (b) `DCoM-Multi` models.

## 5  TRAINING, EVALUATION AND INFERENCE

We trained our model on the train dataset, validated it on the validation dataset and finally reported our results on the test dataset. As class imbalance is present, like Hulsebos et al. (2019), we also evaluated our model performance using the average F1-score, weighted by the number of columns per class in the test set (i.e., the support).

The DCoM models are trained in Tensorflow 2 (Abadi et al., 2015) and the hyperparameters are tuned using `keras-tuner` (O'Malley et al., 2019). For inputs of DCoM models, we tried permutations with replacement as well as without replacement, but we did not observe any significant difference in outputs. The value of $r$ for which we observed best performance for `DCoM-Multi` models is 45. For tokenizing text inputs, we experimented with character-based, word-based and BERT Wordpiece (Wu et al., 2016) tokenizers, and we found out BERT Wordpiece tokenizer stood out to be working better with respect to the other tokenizers because of the obvious reasons stated in the paper (Wu et al., 2016). We compared our result with and without pre-trained embedding weights. It is observed that, though in the final output there is no considerable difference, training with pre-trained embedding weights take approximately 40% less amount of time to converge. We experimented with the small, base and large variations of several BERT architectures and finalized `DistilBERT-base` (Sanh et al., 2019) and `Electra-small` (Clark et al., 2020) based on their performances. We used Bi-directional LSTM layers in all the `DCoM-LSTM` variants. For `Dropout` layers we used 0.3 as our dropout rate. We experimented `mean`, `sum`, `concatenation` and `weighted sum` functions for aggregation, but there was not any significant difference in outputs based on these variations. We used `Adam` (Kingma & Ba, 2014) optimizer with initial learning rate $10^{-4}$. Along with this, we implement a learning rate reducer with a factor of 0.5, which reduces the learning rate by 50% if the model performance does not improve on the validation dataset after consecutive 5 epochs. Assigning class weights does not have much effect on the test results. As this is a multi-class (78 classes) classification problem, the DCoM models are trained with `categorical-crossentropy` loss and validated with `accuracy` and `average F1` score metric.

We tried two approaches for inference on the test dataset. In the first approach, we performed single time inference on each of the instances of the test dataset. While doing so, we set $r$ to be the total number of values ($n_i$) for each instance for `DCoM-Single` models to perform inference. For `DCoM-Multi`, as $r$ values if prefixed, some values will be truncated for inputs with total values $n > r$. The other approach is to generate $k$ (where, $k > 1$) instances with permutation, sampling $r$ values $k$ times between 1 and $n_i$, where $n_i$ is the total number of values for instance $i$. Therefore, we get $k$ class predictions for each of the instances and finally with majority voting we pick the prediction class for each instance. We used $k$ value to be 10 in our case and it is observed that for both `DCoM-Single` and `DCoM-Multi`, we observed $0.2 - 0.5\%$ improvement in the test `average F1` score, but this improvement comes with the price of increased inference time by approximately $k$ times.

## 6 RESULTS

We compared DCoM against Sherlock and other types of models shown by Hulsebos et al. (2019) along with SATO (Zhang et al., 2019), TURL (Deng et al., 2020), TABBIE (Iida et al., 2021), DODUO (Suhara et al., 2021) and TCN (Wang et al., 2021) assuming those to be our baseline models. All the models are compared on the dataset used by Hulsebos et al. (2019) in Sherlock paper. Table 2 presents the comparison of results on the test dataset for each of the models. Columns `Engineered Features` and `k` are specific to DCoM models. `Engineered Features` says whether the 19 engineered features are used with the text inputs while training the model. `k` column tells the number of times inference is performed on a single test instance. It is discussed in detail in section 5. `Runtime` column is the average time in seconds needed to extract features and generate a prediction for a single sample, and `Size` column reports the space required by the trained models in MB. From the table, it is seen that several DCoM models outperform all the baseline models in both F1 score and inference run time with significant margins.

Hulsebos et al. (2019) extracted various features e.g. global statistics, character-level distributions, word embeddings, paragraph vectors from the data and used those features to fit a feedforward neural network model. On the other hand, we treated the data as texts and feed those to more advanced NLP-based models. This allows the DCoM models to extract and learn more useful features that Hulsebos et al. (2019) were unable to extract using hand-engineering. Along with this, hand-engineering takes a considerable amount of time to calculate the features which increase the inference time of Sherlock (Hulsebos et al., 2019) by 3 to 20 times with respect to DCoM models.

| Method | Engineered Features | k | F1 Score | Runtime (s) | Size (MB) |
|---|---|---|---|---|---|
| DCoM-Single-LSTM | Yes | 1 | 0.895 | $0.019 \pm 0.01$ | 112.1 |
| DCoM-Single-LSTM | Yes | 10 | 0.898 | $0.152 \pm 0.01$ | 112.1 |
| DCoM-Single-LSTM | No | 1 | 0.871 | $0.018 \pm 0.01$ | 97.8 |
| DCoM-Single-LSTM | No | 10 | 0.877 | $0.141 \pm 0.01$ | 97.8 |
| DCoM-Multi-LSTM | Yes | 1 | 0.878 | $0.046 \pm 0.01$ | 4.7 |
| DCoM-Multi-LSTM | Yes | 10 | 0.881 | $0.416 \pm 0.01$ | 4.7 |
| DCoM-Multi-LSTM | No | 1 | 0.869 | $0.044 \pm 0.01$ | 4.6 |
| DCoM-Multi-LSTM | No | 10 | 0.871 | $0.401 \pm 0.01$ | 4.6 |
| DCoM-Single-DistilBERT | Yes | 1 | **0.922** | $0.162 \pm 0.01$ | 268.2 |
| DCoM-Single-DistilBERT | Yes | 10 | **0.925** | $1.552 \pm 0.01$ | 268.2 |
| DCoM-Single-DistilBERT | No | 1 | 0.901 | $0.158 \pm 0.01$ | 202.3 |
| DCoM-Single-DistilBERT | No | 10 | **0.904** | $1.492 \pm 0.01$ | 202.3 |
| DCoM-Single-Electra | Yes | 1 | **0.907** | $0.093 \pm 0.01$ | 53.1 |
| DCoM-Single-Electra | Yes | 10 | **0.909** | $0.894 \pm 0.01$ | 53.1 |
| DCoM-Single-Electra | No | 1 | 0.890 | $0.092 \pm 0.01$ | 45.7 |
| DCoM-Single-Electra | No | 10 | 0.892 | $0.887 \pm 0.01$ | 45.7 |
| SATO (Zhang et al., 2019) | - | - | 0.891 | $0.51 \pm 0.01$ | 25.7 |
| TABBIE (Iida et al., 2021) | - | - | 0.897 | $0.058 \pm 0.01$ | 613.4 |
| DODUO (Suhara et al., 2021) | - | - | 0.904 | $0.020 \pm 0.01$ | 155.8 |
| TCN (Wang et al., 2021) | - | - | 0.897 | $0.017 \pm 0.01$ | 108.7 |
| TURL (Deng et al., 2020) | - | - | 0.894 | $0.017 \pm 0.01$ | 110.2 |
| Sherlock (Hulsebos et al., 2019) | - | - | 0.890 | $0.42 \pm 0.01$ | 6.2 |
| Decision tree (Hulsebos et al., 2019) | - | - | 0.760 | $0.26 \pm 0.01$ | 59.1 |
| Random Forest (Hulsebos et al., 2019) | - | - | 0.840 | $0.26 \pm 0.01$ | 760.4 |
| Dictionary (Hulsebos et al., 2019) | - | - | 0.160 | $0.01 \pm 0.03$ | 0.5 |
| Regular expression (Hulsebos et al., 2019) | - | - | 0.040 | $0.01 \pm 0.03$ | 0.01 |
| Consensus (Hulsebos et al., 2019) | - | - | 0.320 | $33.74 \pm 0.86$ | - |

Table 2: Support-weighted F1 score, runtime at prediction, and size of DCoM and other benchmarks

| | Type | F1 Score | Precision | Recall | Support |
|---|---|---|---|---|---|
| | ISBN | 0.988 | 0.999 | 0.993 | 1430 |
| | Grades | 0.988 | 0.995 | 0.992 | 1765 |
| Top 5 Types | Birth Date | 0.987 | 0.985 | 0.986 | 479 |
| | Jockey | 0.983 | 0.988 | 0.986 | 2817 |
| | Industry | 0.984 | 0.984 | 0.984 | 2958 |
| | Person | 0.813 | 0.596 | 0.688 | 579 |
| | Rank | 0.586 | 0.764 | 0.663 | 2978 |
| Bottom 5 Types | Director | 0.702 | 0.549 | 0.616 | 225 |
| | Sales | 0.667 | 0.435 | 0.526 | 322 |
| | Ranking | 0.695 | 0.348 | 0.464 | 439 |

Table 3: Top and bottom five types by F1 score on the test dataset for `DCoM-Single-DistilBERT` model variant.

| | Examples | True Type | Predicted Type |
|---|---|---|---|
| | 1, 5, 4, 3, 2 | Day | Rank |
| Low Precision | 1, 2, 3, 4, 5, 6, 7, 8, 9, 10 | Region | Rank |
| | 1, 2, 3 | Position | Rank |
| | 41, 2, 36 | Ranking | Rank |
| Low Recall | 0, 2, 4 | Ranking | Plays |
| | 1, 2, 3, 4, 5, 6, 7, 8, 9, 10 | Ranking | Rank |

Table 4: Examples of low precision and low recall types on the test dataset for `DCoM-Single-DistilBERT` model variant.

## 6.1 PERFORMANCE FOR INDIVIDUAL TYPES

Following Hulsebos et al. (2019), we also prepared Table 3 which displays the top and bottom five types, as measured by the F1 score achieved by the best performing DCoM model, `DCoM-Single-DistilBERT` for single inference ($k = 1$) for that type. . High performing classes such as grades, industry, ISBN, etc. contain a finite set of valid values which help DCoM models to extract distinctive features with respect to the other classes. To understand types for which `DCoM-Single-DistilBERT` performs poorly, we include incorrectly predicted examples for the lowest precision type (Rank) and the lowest recall type (Ranking) in Table 4. From the table, it is observed that purely numerical values or values appearing in multiple classes, cause a challenge for the DCoM models to correctly classify the semantic type of the data. This issue is discussed in detail in section 7.

## 6.2 FEATURE IMPORTANCE

To calculate the feature importance of 19 engineered features used in trained `DCoM-Single-DistilBERT` model, we extracted the learned feature weight matrix, $W$ from the `dense` layer used after the engineered features input, where $W \in R^{19 \times D}$. Here $D$ is the number of units used in the `dense` layer. $W$ contains the weights/contribution of each of the 19 features on each of the $D$ units, thus forming a $19 \times D$ array, which contains both positive and negative values based on the direction of contribution. We take the absolute value of all the elements of $W$, as we are only interested in the amount of contribution of each of the engineered features, not the direction. After that, we take the mean across the 19 features that result in a 19-dimensional array. We normalized the array by dividing the maximum value of the array by each of the elements. Table 5 enlists the importance of the engineered features in decreasing order. The feature importance score of the engineered features changes with the different variants of DCoM models but the rank of the top 10 features remain intact.

## 7 KNOWN LIMITATIONS

The major limitation of the process is with the pre-processed data prepared by Hulsebos et al. (2019) used to train DCoM models. The same values of instances are present in multiple classes, thus

| Rank | Aux Features | Score |
|---|---|---|
| 1 | Std of # of Numeric Characters in Cells | 1.00 |
| 2 | Std of # of Alphabetic Characters in Cells | 0.63 |
| 3 | Entropy | 0.63 |
| 4 | Std of # of Special Characters in Cells | 0.62 |
| 5 | Std of # of Words in Cells | 0.53 |
| 6 | Mean # Words in Cells | 0.50 |
| 7 | Mean # of Numeric Characters in Cells | 0.48 |
| 8 | Minimum Value Length | 0.45 |
| 9 | Kurtosis of the Length of Values | 0.41 |
| 10 | Mean # Special Characters in Cells | 0.39 |
| 11 | Number of Values | 0.34 |
| 12 | Fraction of Cells with Alphabetical Characters | 0.33 |
| 13 | Fraction of Cells with Numeric Characters | 0.31 |
| 14 | Sum of the Length of Values | 0.31 |
| 15 | Maximum Value Length | 0.30 |
| 16 | Skewness of the Length of Values | 0.28 |
| 17 | Mean # Alphabetic Characters in Cells | 0.28 |
| 18 | Median Length of Values | 0.27 |
| 19 | Mode Length of Values | 0.20 |

Table 5: Feature importance of 19 engineered features used in `DCoM-Single-DistilBERT` model.

| Values | Class |
|---|---|
| F, M | gender |
| M | sex |
| M | gender |
| M, F | sex |
| 1, 2, 3, 4, 5, 6, 7, 8, 9, 10, 11, 12, 13, 14 | ranking |
| 1, 2, 3, 4, 5, 6, 7, 8, 9, 10, 11, 12, 13, 14 | position |
| 1, 2, 3, 4, 5, 6, 7, 8, 9, 10, 11, 12, 13, 14 | rank |
| Fresno State, Oregon, UCLA, South Carolina | team |
| Michigan, Indiana, Wisconsin, Purdue | team name |

Table 6: A sample example of class overlapping of the data used to build up models

resulting in strong class overlapping among some classes. It makes the models confusing during training to learn distinctive features for proper classification. Therefore, for some classes in the test dataset, the models perform very poorly. Table 6 presents a sample of examples of class overlapping in the data prepared by Hulsebos et al. (2019). It is to be noted that the proportion of this class overlap is considerably large in numbers with respect to the total number of instances. Therefore, it affects the training as well as the model performance on the test dataset by a significant margin. Besides the class overlap, faulty or wrong values are present in some classes.

From the application point of view of semantic data type detection models, identifying and preparing a good dataset for training is very challenging as well as time-consuming. Non-standardized column names are a major challenge large organizations face in doing semantic detection. For example, `social security numbers` can be called as `ssn, ssn_id, soc_sec_bnr` etc. Non-standardized data columns also pose a major challenge in information ambiguity. For example, `Gender` column can have values `Male, Female, unknown`, or `0,1,2`. Some organization can have mixed attributes which poses a major challenge in data security. For example, use of PII data like `Social security Number` in `Customer ID` columns. Some other challenges like corrupt/missing metadata also exist.

## 8 CONCLUSION

DCoM presents a novel permutation-based method by which the instances can be fed to the deep learning models directly as natural language. This takes the leverage of using more advanced NLP-based layers/models unlike feedforward neural networks in Sherlock (Hulsebos et al., 2019). The permutation-based method also helps in generating a large number of new instances from the existing ones, helping DCoM models to boost their performance effectively, thus outperforming Sherlock (Hulsebos et al., 2019) and other recent works by quite a significant margin in both F1 scores as well as inference time. We also present an ensembling approach during inference time which improves the test `average F1` score by $0.2 - 0.5\%$.

As mentioned earlier, the next step of it to introduce the context of the column values to DCoM models while predicting the semantic types in relational tables.

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

# A  DATASET DETAILS

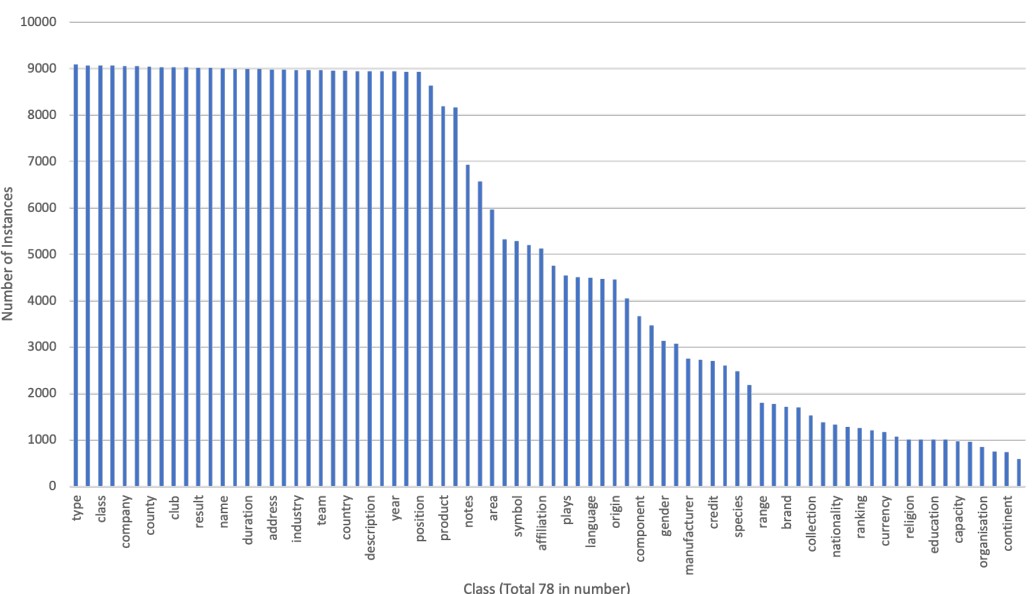

Figure 2: Count of instances for each of the 78 classes in the dataset Hulsebos et al. (2019).

| Instance | Class |
|---|---|
| 1, 2, 3, 4, 5, 6, 7, 8 | day |
| 4:59, 2:44, 2:04, 2:05, 1:13, 3:14 | duration |
| Education, Poverty, Unemployment, Employment | category |
| c;, end-code, code name, label name | command |
| 31 years, 22 years, –, 39 years, 24 years | age |
| 1, 2 | position |
| LA, CA, AL, Warwickshire] | state |
| Deletes the property, Lets you edit the value of the property, Script execution will be stopped | description |

Table 7: A sample of the dataset.

