# OpenReview forum: "DCoM: A Deep Column Mapper for Semantic Data Type Detection"
_ICLR.cc/2022/Conference — ICLR 2022 Submitted_

### Official Review · Reviewer_S4in · 2021-10-29

**Correctness:** 3
**Technical Novelty And Significance:** 2
**Empirical Novelty And Significance:** 2
**Recommendation:** 6
**Confidence:** 4

**Main Review:**

Main Review

The paper studies a very practical problem called semantic type detection. The problem is fundamental to tabular data that consists of columns and headers. Given the widespread adoption of tabular data in database systems, the paper has the potential to benefit extensive applications. The paper presents a new method DCoM for the problem. DCoM avoids the feature engineering step, which is common in existing methods. Also, DCoM achieves satisfactory performance that hits a support-weighted F1 as high as 0.925. However, the paper can be more convincing if several aspects can be taken into consideration. Details are as the following:

1.	Pretrained models should be included as baselines. The proposed DCoM adopts a pre-trained model in the design. For example, DCoM-Single-DistillBERT leverages the pre-trained model DistillBERT that comes from HuggingFace Transformers library. The library contains other pre-trained models, including BERT, RoBERTa, and ALBERT. The interface of using these models is the same as DistillBERT. It will be interesting to see whether different pre-trained model leads to different performance of DCoM, and more importantly, whether these models standalone can solve the semantic type detection problem adequately. The paper will be more convincing if it can include the support-weighted F1 scores of one or more of these pre-trained models as baselines. For experiments, the semantic type detection problem can be naturally formulated as sequence prediction, which is the strength of pre-trained models.

2.	Ablation study is favorable. DCoM presents impressive performance when compared with existing methods. However, it remains unclear why DCoM can achieve superior performance. Is it because DCoM uses higher-dimensional word vectors or DCoM uses larger neural network? The paper can be more insightful if it conducts some ablation study to find out why DCoM is superior.


3.	Real applications can be discussed further. Semantic type detection seems a fundamental problem in tabular data, and an immediate application is header prediction. However, header prediction may not be common because the headers and columns in a table are usually serialized at the same time. Semantic type detection can be more useful if it can affect many downstream tasks (e.g. column matching and schema matching). Some discussions on applications will be helpful to make DCoM adopted extensively in the industry.


**Summary Of The Paper:**

This paper studies the problem of semantic type detection that takes a column of texts then predicts a header name for the column. Existing methods mostly adopt regular expressions or dictionary-lookup or feature engineering. The paper introduces a new method, called DCoM that is different from existing methods. DCoM adopts deep neural networks to process raw text tokens directly instead of relying on feature engineering. The paper conducts extensive experiments and shows that DCoM outperforms a number of contemporary methods on VizNet dataset, which has 78 types.

**Summary Of The Review:**

Summary of the review
The paper studies a novel problem called semantic type detection, which has potential applications over tabular data. However, the paper should include pre-trained models as baselines to show the proposed DCoM outperforms the state-of-the-art methods by a significant margin.

---

> ### Author Response · Authors · 2021-11-22
> **Brief Response of Queries**
>
> 1. We have compared the performance among other variants of BERT as well e.g. BERT, RoBERTa, ALBERT, etc. and it is observed that the base variants of these models produce the same results and they work better than the small variants. This made us believe that DCoM model performance is dependant on the size of the model. We hope that the large variants of these models can produce better than the SOTA result we reported, but due to GPU constraints, we are unable to perform that experiment.
>
> 2. The ablation is study is shown in Table 3, where we reported the result using different variants of DCoM models. We have tested the effect of input type, engineered features, model size, etc creating multiple variations. Therefore, to answer your question, the reason DCoM performing superior is primarily because of the permutation-based natural language input to the model. Along with that, the model size has also a positive impact on the higher support-weighted F1 score as shown in Table 3, where we can see that as the model size increases from LSTM to BERT-based models performance improves.
>
> 3. DCoM is extensively used in our organization as a tool to detect and mask sensitive columns from the data. This has been discussed in Sections 1 and 7 of the paper.

---

> > ### Comment · Reviewer_S4in · 2021-11-24
> > **Thank you for the response!**
> >
> > Thank you for the author response! It is nice to see that the authors have conducted additional experiments and discussed the ablation study as well as real use cases. Please consider to include new results and discussions in the paper.
> >
> > If the paper is not accepted at this conference after reviewers' discussion,  I encourage the authors to state quantitatively (e.g. XX%) the F1 score improvements of DCoM over existing. Also, it will be an interesting problem that whether the proposed technique, i.e. permutation-based natural language input, is generally helpful to all BERT tasks, such as sentence-pair classification and question answering. If so, the paper will create broader impact than its current scope which focuses on semantic type detection task only.

---

### Official Review · Reviewer_MWJi · 2021-10-30

**Correctness:** 2
**Technical Novelty And Significance:** 1
**Empirical Novelty And Significance:** 1
**Recommendation:** 3
**Confidence:** 5

**Main Review:**

### Strength

- (S1) The paper combines learned representation and manually crafted features for further improvements on the semantic detection task.


### Weaknesses

- (W1) The novelty and technical significance are limited.
- (W2) Missing important references and comparison against the state-of-the-art methods.
- (W3) DCoM-Multi does not show improvements over DCoM-Single.

### Major comments

(W1)

Using raw table values for semantic type detection is a commonly used approach and several models have been already developed [1-4]. The models in [1-3] also incorporate table structures (i.e., row/column) into the model, which is not considered in DCoM.

- [1] Xiang Deng, Huan Sun, Alyssa Lees, You Wu, Cong Yu, TURL: Table Understanding through Representation Learning, VLDB 2021 (https://arxiv.org/abs/2006.14806)
- [2] Hiroshi Iida, Dung Thai, Varun Manjunatha, Mohit Iyyer, TABBIE: Pretrained Representations of Tabular Data, NAACL 2021 (https://arxiv.org/abs/2105.02584)
- [3] Daheng Wang, Prashant Shiralkar, Colin Lockard, Binxuan Huang, Xin Luna Dong, Meng Jiang, TCN: Table Convolutional Network for Web Table Interpretation, WebConf 2021. (https://arxiv.org/abs/2102.09460)
- [4] Yoshihiko Suhara, Jinfeng Li, Yuliang Li, Dan Zhang, Çağatay Demiralp, Chen Chen, Wang-Chiew Tan, Annotating Columns with Pre-trained Language Models, arXiv April 2021 (https://arxiv.org/abs/2104.01785)

Furthermore, all the models and Sato [Zhang et al. (2019)] are multi-column models that consider all columns in the same table to detect the semantic types in a joint manner. As discussed in [Zhang et al. (2019)], the table context is essential to semantic type detection.


(W2)

As commented above, the paper should cite and compare with [1-4] in addition to Sato, which is cited but not compared in the paper.


(W3)

According to Table 3, DCoM-Multi does not show improvements over DCoM-Single.


### Minor comments

- Why did the author(s) use the 19 features out of 27 Sherlock features?
- Do the feature importance scores for different models (e.g., DistilBERT) follow the same or a different trend against the feature importance scores reported in Table 6?
- Typo: we -> We (p.5 2nd paragraph)


**Summary Of The Paper:**

This paper develops a semantic type detection model DCoM that takes column values as text in addition to auxiliary features extracted by the exiting semantic type detection method Sherlock. The paper proposes two versions: (1) Single, which takes a single sequence, and (2) Multi, which takes multiple sequences using permutation. Experimental results on the VizNet corpus show that DCoM models with DistilBERT/ELECTRA outperform Sherlock and other non-Deep Learning models.

**Summary Of The Review:**

The paper has critical issues with respect to novelty and technical significance. The paper should cite the missing papers and compare with those models (and Sato.) Thus, I wouldn’t recommend publishing the paper in its current form.

---

> ### Author Response · Authors · 2021-11-22
> **Brief Responses on Comments**
>
> 1. I have cited the above-mentioned five references and compared the performance of those models with DCoM on the VizNet corpus dataset. The revised paper has been submitted.
>
> 2. DCoM-multi does not show improvements over DCoM-Single. As DCoM-Single feeds permutation-based sequential input to the model for a single instance we can provide a large number of column values to the model at a time. On the other hand, DCoM-multi feeds multiple column values in parallel. The bottleneck is If we increase the number of parallel inputs it increases the number of parameters of the model and due to space constraints we were unable to experiment with a large number of parallel inputs (around 10). As one parallel input contains one column value, that total number of column values we could feed to the model in DCoM-Multi is very lesser than DCoM-Single.  This may be one of the reasons of DCoM-Multi resulting worse than DCoM-Single.
>
> 3. We used 19 out of 27 Sherlock features. The rest 8 features do not make sense as we were considering the unique values of the columns e.g. one of the left out features is "Fraction of values with unique content".
>
> 4. The feature importance score of the engineered features changes with the different variants of DCoM models but the rank of the top 10 features remain intact. We have added this line to the paper.

---

> > ### Comment · Reviewer_MWJi · 2021-11-29
> > **Reply to the author response**
> >
> > Thank you for the author response.
> >
> > > 1
> > Table 2 in the revised version shows inconsistent results compared to the results in the Sato (Zhang et al. 2019), TURL (Deng et al. 2020), and Doduo (Suhara et al. 2021) papers. It is shown that
> >
> > - Sato >> Sherlock (90.2 vs 86.7 F1 on the VizNet dataset) in (Zhang et al. 2019)
> > - Doduo >> Sato (94.3 vs 88.4 F1 on the VizNet dataset) in (Suhara et al. 2021)
> > - TURL >> Sherlock (88.86* vs 78.47 F1 on the WikiTable dataset) in TURL (Deng et al. 2020)
> > (*) TURL w/o metadata, assuming that this paper doesn’t consider/use any metadata
> >
> > However, the performance differences between those methods look marginal in Table 2. This looks strange to me because (Zhang et al. 2019) and (Suhara et al. 2021) use the same VizNet dataset as this paper (Technically speaking, the Sherlock version and the Sato version are different but are created from the same VizNet corpus.) Especially, the marginal difference between Sato (0.891) and Sherlock (0.890) makes me worry about the experimental settings. Sato is an extended version of Sherlock, which uses additional LDA features in addition to a CRF layer for structured output prediction. Thus, it should outperform Sherlock in general (and it is shown in the Sato paper, which used the same VizNet corpus.)
> >
> > Perhaps, the dataset was used in a column-wise prediction manner (i.e., decomposing tables into individual columns) instead of using the original table as input. This explains the marginal differences between the methods (including Sato vs Sherlock)
> >
> > If this is the case, the experimental settings are unnecessarily unfair to those methods and unrealistic, as the entire table should be considered for column type annotation (as done in (Zhang et al. 2019), (Deng et al. 2020), (Wang et al. 2020), (Iida et al. 2021), (Suhara et al. 2021))

---

### Official Review · Reviewer_S7C6 · 2021-11-03

**Correctness:** 2
**Technical Novelty And Significance:** 2
**Empirical Novelty And Significance:** 2
**Recommendation:** 3
**Confidence:** 4

**Main Review:**

Pros:
- The task is interesting/important
- Good performance is reported

Cons:
- While the the context information of the column values is of interest, modeling this information is deferred to future work.
- Classifying the semantic type based only on the string value is too simplistic. One of the main challenges of this task in practice is to resolve ambiguities, where the authors do not address this aspect altogether.
- Assuming that the classifier/s may just memorize values per category, the authors should check/report the overlap between train and test.
- Otherwise, it would be interesting to train and test on different datasets, to assess learning generalization.
- I'm not sure that the dataset contains enough diverse examples for learning 70+ categories.

Some typos:
- many attempts by Microsoft (2019): cannot cite like this.. (repeats several times)
- We refrained ourselves extracting --> refrained from
- The output of the same is aggregated --> ??
- Figure 2 uses a too small font.


**Summary Of The Paper:**

The goal of the work is to associate semantic data types to table attributes. This task is of importance to data cleaning, schema matching,  etc. The authors claim that existing methods use regular expressions or dictionary lookups, and proposed a learning approach to the problem.
They feed raw values (strings) to neural classifiers, and train their models on 686,765 data columns extracted from the VizNet corpus with 78 different semantic data types. The textual values processed by LSTM/Transformer/BERT layers, where in addition the authors use 19 tatistical features from Hulsebos et al. (2019).  Overall, they achieve F1 score of 0.925.

Analysis showed that high performance is obtained for classes that contain a finite set of valid values, e.g. grades or industry.
On the other hand, numerical values or in general values that appear in multiple classes, are more challenging to classify.


**Summary Of The Review:**

The approach is overly simplistic, as it does not account for ambiguities. Methodologically, I'm not convinced that the model generalizes, as there may be significant overlap between the train and test examples.

---

> ### Author Response · Authors · 2021-11-22
> **Responses on the Comments**
>
> 1. The dataset has been benchmarked by the authors of Sherlock. Therefore we did not have much flexibility to alter its distribution. The points you raised are very valid and the same we have discussed briefly in Section 7.
>
> 2. We have rectified the typographical errors and submitted the revised copy.

---

### Decision · Program_Chairs · 2022-01-20

**Decision:**

Reject

**Comment:**

The paper studies semantic type detection.
 The problem is of practical significance to  i  tabular data.
 However, in its current form, there are concerns about  the scope of novelty and technical significance.